# Neutrophil Count as Atrioventricular Block (AVB) Predictor following Pediatric Heart Surgery

**DOI:** 10.3390/ijms232012409

**Published:** 2022-10-17

**Authors:** Tomasz Urbanowicz, Anna Olasińska-Wiśniewska, Marcin Gładki, Michał Michalak, Mateusz Sochacki, Anita Weclewska, Dominika Zalas, Waldemar Bobkowski, Marek Jemielity

**Affiliations:** 1Cardiac Surgery and Transplantology Department, Poznan University of Medical Sciences, 61-848 Poznan, Poland; 2Pediatric Cardiac Surgery Department, Poznan University of Medical Sciences, 60-572 Poznan, Poland; 3Department of Computer Science and Statistics, Poznan University of Medical Sciences, 60-806 Poznan, Poland; 4Poznan University of Medical Sciences, 61-701 Poznan, Poland; 5Pediatric Cardiology Department, Poznan University of Medical Sciences, 60-572 Poznan, Poland

**Keywords:** neutrophil 1, cardiopulmonary bypass 2, AVB 3, pediatric heart surgery 4, NLR 5, SIRI 6, SII 7

## Abstract

Neutrophils play a significant role in immune and inflammatory reactions. The preoperative inflammatory activation may have a detrimental effect on postoperative outcomes. The aim of the study was to investigate the relation between preoperative hematological indices on postoperative complications’ risk in pediatric cardiac congenital surgery. The retrospective single center analysis included 93 pediatric patients (48 (65%) males and 45 (35%) females), mean age of 7 (3–30) months referred for cardiac surgery in cardiopulmonary bypass due to functional single ventricle disease (26 procedures), shunts lesions (40 procedures) and cyanotic disease (27 procedures). Among simple hematological indices, the receiver-operating-characteristic curve showed that a neutrophil count below 2.59 K/uL was found as an optimal cut-off point for predicting postoperative atrioventricular block following pediatric cardiac surgery (AUC = 0.845, *p* < 0.0001) yielding a sensitivity of 100% and a specificity of 65.62%. Preoperative values of neutrophil count below 2.59 K/uL in whole blood analysis can be regarded as a predictive factor (AUC = 0.845, *p* < 0.0001) for postoperative atrioventricular block in pediatric cardiac surgery.

## 1. Introduction

The overall risk for complication in pediatric heart surgery is reported to be as high as 40% [1]. The Risk Adjustment for Congenital Heart Surgery (RACHS-1) method was established to evaluate the perioperative risk, as reported major complications in cardiac surgery is 13% [2,3]. One such complication is perioperative atroventricular block (AVB) with the incidence of 1.6% as reported by Paech et al. [4].

Several inherited and acquired factors may interfere with proper heart function in infancy. Some of them co-exist impeding the interpretation. Cells and tissue damage and continuous repair processes allow normal circulatory activity. Among other blood cells, neutrophils play a crucial role in the healing processes of cardiovascular system, with the repair being driven mainly by angiogenesis [5,6]. Their role in conduction disturbances has been postulated [7]. Although initially believed to be involved only in immune defense, there is growing evidence that neutrophils belong to the group of recruited cells in healing and repairs as presented in animal models [8].

The cardiac surgical intervention, especially including cardiopulmonary bypass application, triggers inflammatory activation and is related to some extent of organ dysfunction [9,10,11,12].

The aim of the study was to investigate the relation between preoperative whole blood count components and postoperative complications’ risk in pediatric cardiac surgical procedures.

## 2. Results

There were 93 patients (median age: 7 (3–30) months), who underwent cardiac surgery between July 2020 and Ferbruary 2021 in the Deparment of Pediatric Cardiac Surgery in Poznan University of Medical Sciences, enrolled into the study. The following procedures were performed in this group: 26 surgeries in children with functional single ventricle diagnosis, 27 surgeries in children with other cyanotic diseases and 40 shunt lesions, as presented in Figure 1.

Detailed intraoperative and postoperative information is presented in Table 1.

The postoperative heart rhythm disturbances turned out to be associated with preoperative blood morphology results. The neutrophils count was significantly decreased in patients with postoperative atrioventricular 2nd and 3rd degree AVB (*p* = 0.009). The neutrophil count did not, however, differ between patients with and without pacemaker implantation. On the contrary, tachyarrhythmias in the postoperative period were more common in patients with a higher leucocyte count (*p* = 0.017).

The logistic regression analysis of preoperative factors including clinical data and inflammatory indices from whole blood count analysis and postoperative complications was performed. The relation between the preoperative neutrophil count and the risk for a permanent pacemaker was revealed, as presented in Table 2.

Univariate and multivariate analyzes of age, sex, type of disease, and laboratory results, including troponin levels and neutrophils components of whole blood count analysis, did not present any mortality significance in the studied group. The analysis focused on other complications, including kidney failure requiring hemofiltration, prolonged intubation and multiorgan failure, and milrinone therapy was performed, as presented in Table 3.

There was 3rd degree AV block diagnosed in four patients undergoing heart surgery and 2nd degree AV block in one patient requiring pacing. The three of them required permanent pacemaker implantation on the 8th, 10th, 11th day following surgery. In two more, the conduction disturbances were transient and resolved on the 6th and 8th postoperative days. The receiver-operating characteristics showed that a neutrophil count below 2.59 K/uL was found as the optimal cut-off point for the prediction of perioperative AVB following pediatric cardiac surgery (AUC = 0.845, *p* < 0.0001) yielding a sensitivity of 100% and a specificity of 65.62%, as presented in Figure 2.

The two-year survival was assessed based on the cut-off value of the neutrophils predictive for AVB occurrence; however, the Kaplan–Meier analysis did not reveal significant differences in the long-term survival outcome between patients with baseline neutrophils over and below 2.59 K/uL (96.4% and 88.6%, respectively, *p* = 0.137) as presented in Figure 3.

## 3. Discussion

This is the first, to our best knowledge, analysis presenting the relation between the preoperative peripheral blood neutrophil count and the risk for postoperative atrioventricular rhythm disturbances following cardiac surgery in children. The second finding, increased leukocytosis related to tachyarrhythmias, confirms the previously reported observation presenting the relation between the neutrophil-to-lymphocyte ratio (NLR) and supraventricular arrhythmias [13].

The frequency of complete heart block after pediatric congenital heart surgery is reported in 1% of procedures [14]. The disturbances of the conduction system following heart surgery are observed as late complications due to scare formations in heart tissue [15]. The complete AVB following cardiac surgery in children is fairly common, yet the lack of resolution within seven postoperative days should be a warrant for permanent pacemaker (PPM) implantation. Romer et al. presented that the majority of analyzed patients (94%) had resolution of transient AVB by 10 days after the surgery, so there was limited benefit to delaying the implantation of PPM by more than 10 days postoperatively [16].

The types of surgery related to the highest risk for complete AV block include double switch operation, mitral and tricuspid valve surgery followed by ventricular septal defect (VSD) repair [17].

The immunological background of inherited AVB includes an autoimmune process affecting heart development [18]. In some cases, the maternal autoantibodies can be detected in childhood and even in adults, and induce damage of the heart conduction system [19].

In our analysis, we found the low neutrophil count in peripheral blood as a predictive factor for perioperative atrioventricular block. Neutropenia is common in the children population and attributable to alloimmune or iso- and autoimmunological mechanisms [20]. Alloimmune ethology of neutropenia is related to incompatibility of maternal—fetal antigens [21]. The transplacental transfer of pre-existing maternal IG antibodies are the causative agents for developing the iso-immune neutrophil count derangements in infants [22]. Autoantibodies against neutrophils are responsible for auto-immune neutropenia development [23].

In our analysis, we found the relation between neutrophil count in peripheral blood and as a predictive factor for perioperative AVB that may suggest the relation between inflammatory activations and conducting heart system disturbances. The lower neutrophil count was prognostic for permanent pacemaker implantation. As the different stages of neutropenia are reported in autoimmune diseases, we state that the increased risk for AV block in the presented group could be related to unrecognized/some underling autoimmune disturbances [24] not necessarily diagnosed as autoimmune disease. The possible explanation that may explain the diminished neutrophil count in peripheral blood and secondary increased risk for PPM requirement may also relate to maternal/host autoantibodies.

The results of our analysis not only indicate neutrophil count as a simple and easily obtained marker for prolonged heart conduction disturbances but may suggest the relation between inflammatory derangements and juvenile heart function.

The large-scale reports presented the risk for postoperative AVB, yet in many situations, an explanation for this complication cannot be established, excluding procedures related to heart conductance system anatomy [12].

In autoimmune AVB, some authors described the role of maternal autoantibodies (anti-Ro, anti-La) in triggering cascade that destroys the atrioventricular node [25]. The immunological activation after heart surgery was reported in autoimmune Miller Fisher syndrome [26].

The cardiopulmonary bypass (CPB) application, intraoperative ischemia, followed by reperfusion injury are claimed to trigger inflammatory response. The association between CPB and brain injury in newborns related to systemic inflammation and central nervous system—derived proteins was presented by Pironkowa et al. [27]. The cytokines such as tumor necrosis factor—alpha and interleukins-6 and -8 released by activated inflammatory cells are claimed for complications of the systemic inflammatory respones following heart surgery [28,29,30]. The inflammatory activation in pediatric patients was observed even when ultrafiltration and aprotinin were used in clinical practice as interleukins are released due to reduced oxygen concentration in arterial blood together with increased tissue demands [31,32].

Our analysis is based on a simple marker, easily obtained from whole blood count analysis, which serves as a predictive marker for perioperative significant AB block (2nd or 3rd degree requiring pacing). The relation between preoperative simple inflammatory markers and cardiovascular diseases progression has already been postulated [33,34,35,36]. We focused on preoperative inflammatory indices in children, as previous reports in adults linked postoperative complications to perioperative neutrophil activation [37,38,39,40]. We believe that further research is required to explain the relation between inflammatory response and risk for PPM implantation.

### Study Limitation

This is a single center retrospective analysis presenting the relation between preoperative laboratory results and postoperative complications. Due to the retrospective character of the study, we are unable to evaluate the change in the neutrophil count after surgery. However, we performed analysis of the pre- and post-surgical count of white blood cells and did not find a relation between the count change and PPM implantation. The possible relation between maternal antibodies and neutrophils count, suggested in this discussion, was not available for verification due to the retrospective character of the analysis. The patients operated on with co-existing infections were not included in the analysis.

## 4. Materials and Methods

In all, 93 from 102 consecutive patients referred for pediatric cardiovascular intervention were enrolled into the study. Patients diagnosed with functional singe ventricle (SV), cyanotic disease, and atrial/ventricular heart septum defects (ASD/VSD) were included into the retrospective analysis. Figure 1 presents the composition of the study group.

Patients undergoing isolated valve replacement/repair procedures, procedures without cardiopulmonary bypass (CPB), patients with hematological and oncological diseases were excluded from the study. Detailed information regarding enrolled patients is presented in Table 4.

The study was performed according to the principles of Good Clinical Practice and the Declaration of Helsinki and was approved by the Local Ethics Committee of the Poznan University of Medical Sciences, Poznan, Poland (approval number: 545/22 on 9 June 2022).

### 4.1. Biochemical Parameters

Whole blood count parameters analysis was performed on a routine hematology analyzer (Sysmex Europe GmbH, Norderstedt, Germany). The inflammatory indices were calculated based on the whole blood samples analysis, including white blood count, neutrophils, neutrophil-to-lymphocyte ratio (NLR), systemic immune inflammatory index (SII) (neutrophils x platelets/lymphocyte), and systemic inflammatory response index (SIRI) (neutrophils x monocytes/lymphocyte) [41,42]. Moreover, biochemical analysis included maximum serum troponin and creatinine assessment.

### 4.2. Statistical Analysis

Since data did not follow normal distribution (Shapiro-Wilk test), the parameters were presented as medians and interquartile ranges (Q1–Q3). The categorical data were presented as numbers and percentages. The comparison between groups was performed by the Kruskal–Wallis test with the post hoc Dunn’s test. In case the comparison considered categorical data, the chi-square test of independence was used. A logistics regression analysis was used in order to check if the analyzed parameters are potential risk factors of analyzed complications events. Results were presented as odds ratios (OR) and its 95% confidence intervals (95%CI). Receiver-characteristics-curve (ROC) analysis was performed in order to find predictors of AV block disturbances. The prognostic properties of analyzed predictors were assessed by AUC (area under the curve) An optimal cut-off point was denoted at the highest sensitivity and specificity value. The Kaplan–Meier estimate was used to compare survival curves. The comparison between survival curves was performed by log-rank test. Statistical analysis was performed with the use of MedCalc^®^ Statistical Software version 20.027 (MedCalc Software Ltd., Ostend, Belgium; https://www.medcalc.org; 29 August 2022). All tests were considered significant at *p* < 0.05.

## 5. Conclusions

Preoperative values of neutrophil count below 2.59 K/uL in whole blood analysis can be regarded as a predictive factor (AUC = 0.845, *p* < 0.0001) for postoperative atrioventircular block in pediatric cardiac surgery.

## Figures and Tables

**Figure 1 ijms-23-12409-f001:**
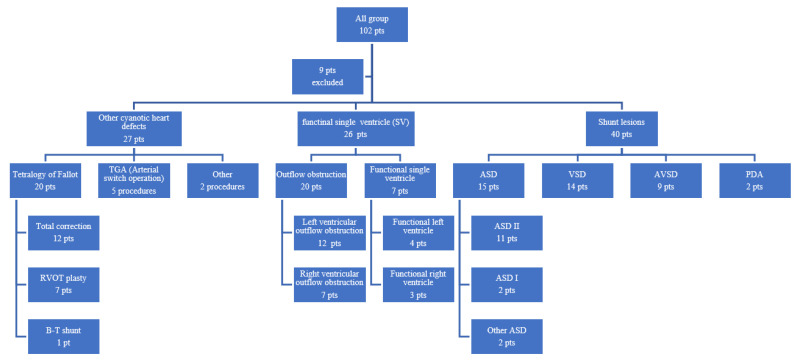
Flow chart. Abbreviations: ASD—atrial septal defect, ASD 1—atrial septal defect type 1, ASD 2—atrial septal defect type 2, AVSD—atrioventricular septal defect, pt—patient, pts—patients, RVOT—right ventricle outflow tract, SV—functional single ventricle.

**Figure 2 ijms-23-12409-f002:**
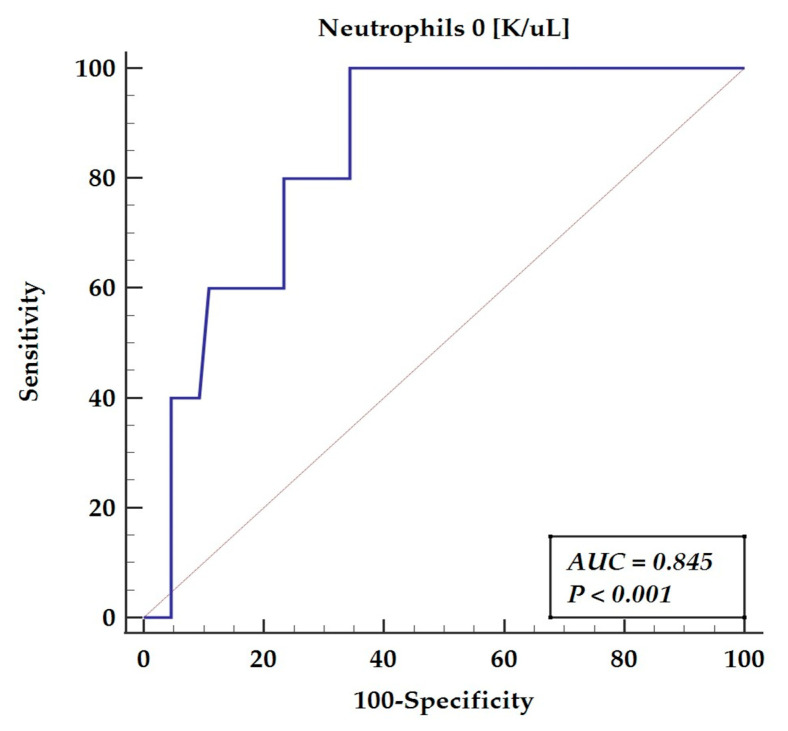
Receiver-operating characteristics for AV block disturbances. Abbreviations: AUC—area under the curve.

**Figure 3 ijms-23-12409-f003:**
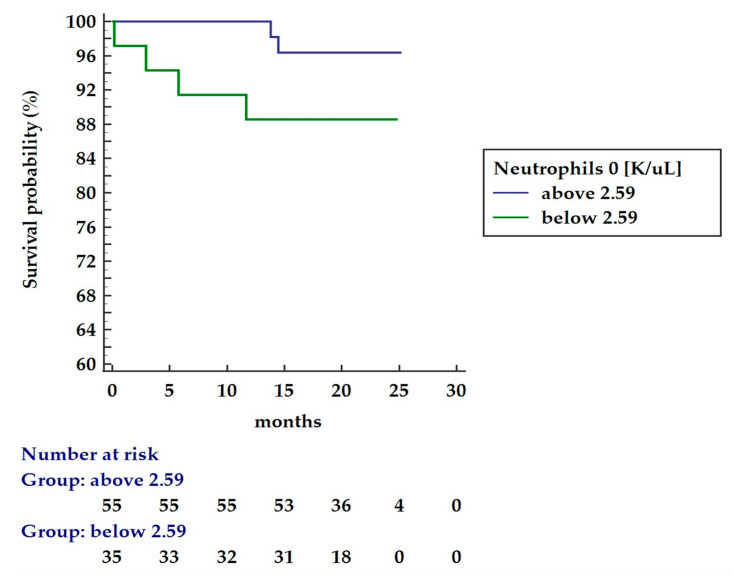
Kaplan–Meier 30 months survival analysis for patients’ cohort based on lower (green) and upper (blue) cut-off (2.59 K/uL) of the neutrophil count.

**Table 1 ijms-23-12409-t001:** Intra- and postoperative results in presented groups.

	All Procedures (N = 93)	SV (N = 26)	Shunt Lesions (N = 40)	Cyanotic (N = 27)	*p*	SV vs. Shunt Lesions	SV vs. Cyanotic	Shunt Lesions vs. Cyanotic
Intraoperative parameters: CBP time Cardioplegia: amount (mL) median (Q1–Q3) Type: Del Nido (%) Calafiore (%) Custodiol (%)	130 (90–170) 88 (94%) 4 (4%) 2 (2%)	96 (76–112) 120 (70–245) 27(100%) 0 0	68 (58–92) 130 (101–165) 37 (92.5%) 1 (2.5%) 2 (5%)	135 (99–163) 130 (100–160) 24 (89%) 3 (11.1%) 0	<0.001 0.886 0.174 0.185 0.169	0.015	0.021	<0.001
Postoperative complications: Inotropes requirement Milrinone infusion Intubation time (h) Intubation > 24 h CVVH ECMO Multiorgan failure AVB	83 75 4 5	23 (85.2%) 21 (77.8%) 30 (8–118) 18 (66.7%) 0 1 (3.7%) 1 (3.7%) 1 (3.7%)	35 (87.5%) 31 (77.5%) 9 (7–56) 18 (45.0%) 1 (2.5%) 1 (2.5%) 1 (2.5%) 2 (5%)	25 (96.2%) 23 (85.2%) 54 (9–127) 17 (62.9%) 2 (7.4%) 1 (3.7%) 3 (11.1%) 2 (7.4%)	0.390 0.345 0.250 0.465 0.154 0.099 0.612 0.895			
Mortality Day of hospitalization		1 (3.7%) 18	1 (2.5%) 5	0 (0%)				
Overall hospitalization (days) median (Q1–Q3)		5 (4–10)	3 (2–5)	7 (4–15)	0.557			

Abbreviations: ASD—atrial septal defect, AVB—atrioventricular block, AVSD—atrioventricular septal defect, CPB—cardiopulmonary bypass, CVVH—continuous veno—venous hemofiltration, ECMO—extracorporeal membrane oxygenation, RVOT—right ventricle outflow tract, SV—functional single ventricle, VSD—ventricular septal defect.

**Table 2 ijms-23-12409-t002:** The logistic regression analysis of preoperative factors for postoperative A-V block.

Parameters	OR	SE	95% CI	*p*
Risk for AVB				
Neutrophil count	0.208	0.164	0.045–0.973	0.046
Lymphocyte count	0.834	0.163	0.569–1.223	0.353
Hb	0.846	0.178	0.489–1.123	0.423
NLR	0.494	0.496	0.069–3.537	0.482
SII	0.998	0.004	0.991–1.005	0.575
SIRI	0.064	0.141	0.001–4.766	0.212
Sex	1.043	0.847	0.212–5.119	0.959
Type of disease - Functional single ventricle - Shunt lesions - Cyanotic				
	1.26	1.131	0.217–7.319	0.797
	0.658	0.587	0.114–3.782	0.639
	1.26	1.131	0.217–7.319	0.797
Age (months)	0.995	0.013	0.971–1.020	0.699
Down syndrome	1.756	2.019	0.184–16.740	0.625
CPB time	0.982	0.016	0.949–1.016	0.295
Cardioplegia	0.419	0.356	0.079–2.216	0.306
Troponin	1.361	1.663	1.302–1.432	0.455

Abbreviations: AVB—atrioventricular block, CPB—cardiopulmonary bypass time, Hb—hemoglobin, NLR—neutrophil-to-lymphocyte ratio, SII—systemic immune inflammatory index, SIRI—systemic inflammatory response index.

**Table 3 ijms-23-12409-t003:** Logistic regression analysis of factors related to postoperative complications.

Parameters	OR	SE	95% CI	*p*
Prolonged intubation (>24 h)				
N	1.179	0.220	0.817–1.700	0.378
L	1.027	0.087	0.871–1.212	0.748
Hb	1.305	0.278	0.879–2.101	0.056
NLR	1.019	0.283	0.592–1.756	0.944
SII	1.000	0.001	0.998–1.002	0.876
SIRI	1.463	0.497	0.752–2.846	0.262
Sex	0.733	0.295	0.333–1.612	0.440
Type of disease HLHS Shunt lesions Cyanotic	1.829 0.444 1.464	0.870 0.189 0.685	0.719–4.647 0.192–1.025 0.585–3.662	0.205 0.057 0.415
Age (months)	0.986	0.006	0.974–0.998	0.023
Down syndrom	1.928	1.396	0.466–7.97	0.365
Cross-clamping time	1.017	0.006	1.001–1.029	0.003
Cardioplegia	0.464	0.218	0.184–1.170	0.104
Pre-troponin	36.32	20.28	0.06–205.54	0.520
Post-troponin	0.992	0.009	0.974–1.011	0.388
Milrinone requirements				
N	1.876	1.094	0.598–5.884	0.281
L	0.991	0.179	0.696–1.411	0.960
Hb	0.997	0.218	0.678–1.325	0.921
NLR	1.408	1.189	0.269–7.371	0.686
SII	1.009	0.001	0.994–1.024	0.236
SIRI	0.702	0.327	0.282–1.748	0.447
Sex	3.186	3.002	0.503–20.197	0.219
Type of disease HLHS Shunt lesions Cyanotic	0.586 0.474 1.658	0.552 0.445 1.021	0.092–3.719 0.075–2.981 0.495–5.547	0.571 0.426 0.411
Age (months)	1.374	0.266	0.939–2.011	0.101
Down syndrome	1.453	1.749	0.074–28.199	0.805
Cross-clamping time	0.997	0.002	0.992–1.003	0.372
Cardioplegia	0.544	0.621	0.058–5.098	0.594
Pre-troponin	9.727	33.842	0.011–889.189	0.513
Post-troponin	1.007	0.0143	0.979–1.036	0.598
Multi organ failure (MOF)				
N	0.731	0.493	0.195–2.741	0.642
L	1.094	0.260	0.687 -1.744	0.705
Hb	1.356	0.271	0.879–3.125	0.062
NLR	0.223	0.513	0.002–20.105	0.514
SII	0.996	0.008	0.981–1.011	0.569
SIRI	0.548	0.917	0.021–14.541	0.719
Sex	3.121	3.628	0.319–30.476	0.328
Type of disease HLHS Shunt lesions Cyanotic	0.821 0.436 2.6	0.966 0.512 2.671	0.082–8.255 0.044–4.353 0.347–19.472	0.867 0.479 0.352
Age (months)	0.905	0.099	0.731–1.121	0.362
Down syndrome	3.0	3.621	0.282–31.952	0.363
Cross-clamping time	1.004	0.006	0.996–1.011	0.255
Cardioplegia	1.354	1.594	0.1349–13.604	0.796
Pre-troponin	60.601	165.117	0.291–126.389	0.132
Post-troponin	0.979	0.034	0.914–1.094	0.558
Kidney failure				
N	1.381	0.369	0.817–2.333	0.228
L	0.976	0.139	0.737–1.292	0.864
Hb	1.378	0.291	0.781–2.781	0.078
NLR	1.286	0.481	0.618–2.675	0.500
SII	1.002	0.001	0.999–1.004	0.150
SIRI	1.438	0.543	0.685–3.016	0.337
Sex	0.529	0.360	0.139–2.009	0.350
Type of disease HLHS Shunt lesions Cyanotic	1.271 0.353 2.157	0.949 0.294 1.539	0.294–5.495 0.069–1.801 2.532–8.738	0.748 0.211 0.282
Age (months)	0.921	0.127	0.702–1.207	0.552
Down syndrome	2.75	2.430	0.487–15.542	0.252
Cross-clamping time	1.014	0.005	1.004–1.024	0.005
Cardioplegia	1.629	1.360	0.317–8.369	0.559
Pre-troponin	26.323	16.482	0.641–195.346	0.567
Post-troponin	0.990	0.019	0.953–1.028	0.617

Abbreviations: HLHS—hypoplastic left heart syndrome, KF—kidney failure, L-lymphocyte count, MOF—multiorgan failure, MR—milrinone requirements, N—neutrophil count, NLR—neutrophil-to-lymphocyte ratio, PI—prolonged intubation (>24 h), PPM—permanent pacemaker, SII—systemic immune inflammatory index, SIRI—systemic inflammatory response index.

**Table 4 ijms-23-12409-t004:** Demographical and clinical characteristics.

	All Group (N = 93)	SV (N = 26)	Shunt Lesions (N = 40)	Cyanotic (N = 27)	*p*	SV vs. Shunt Lesions	SV vs. Cyanotic	Shunt Lesions vs. Cyanotic
Sex (Male/Female)	48/45	11/15	22/18	14/13	0.506			
Weight (kg) Height (cm)	4.9 (2.1–6.4) 69 (61–95)	4.7 (2.2–6.3) 68 (56.5–110)	4.9 (2.0–6.5) 70.5 (61.5 -90.5)	4.8 (2.0–6.5) 70 (61.5–95)	0.950 0.701			
Prenatal factors: 1. FAS 2. prematurity 3. cesarean section	1 42 3	0 14 0	1 12 2	0 16 1	0.505 0.035 0.523	0.055	0.694	0.018
Age (months) median (Q1–Q3)	7 (3–30)	4 (0–30)	7 (4–20)	8 (1–25)	0.910			
Preoperative: Inotropes Intubation Down syndrome Kidney failure Dysmorphia Di George syndrome Duchenne syndrome	3 (3%) 10 (11%) 1 (1%) 3 (3%) 1 (1%) 2 (2.2%) 1 (1%)	0 2 1 (3.7%) 4 (14.8%) 1 (3,7%) 2 (7.7%) 0	0 1 (2,5%) 9 (22.5%) 3 (7.5%) 0 0 1 (2.5%)	3 (11%) 7 (26%) 0 3 (11%) 0 0 1 (3.7%)	0.023 0.007 0.005 0.632 0.285 0.079 0.636	- 0.344 0.040	0.083 0.071 0.308	0.032 0.004 0.009
Preoperative laboratory results: 1. WBC (K/uL) median (Q1–Q3) 2. Neutrophil (K/uL) median (Q1–Q3) 3. Lymphocyte (K/uL) median (Q1–Q3) 4. NLR (median (Q1–Q3) 5. Hb (g/dL) median (Q1–Q3) 6. Platelets (K/uL) median (Q1–Q3) 7. Monocyte (K/uL) median (Q1–Q3) 8. SII median (Q1–Q3) 9. SIRI median (Q1–Q3) 10. creatinine (mg/dL) median (Q1–Q3) 11.Troponin (ng/mL) median (Q1–Q3)	9.6 (8.1–12.6) 2.8 (2.1–3.7) 5.1 (3.4–7.5) 0.4 (0.2–0.8) 12.9 (11.7–14.9) 316 (254–391) 0.7 (0.5–0.8) 160 (115–272) 0.4 (0.3–0.8) 0.3 (0.2–0.4) 0.017 (0.008–0.034)	9.1 (7.2–12.6) 2.7 (2.4–3.7) 3.8 (2.6–4.4) 0.7 (0.5–1.1) 13.6 (11.1–16.1) 266 (220–337) 0.8 (0.6–1.1) 148 (111–235) 0.5 (0.4–0.9) 0.4 (0.3–0.6) 0.017 (0.007–0.035)	10.3 (8.3–11.4) 2.5 (2–3.1) 3.8 (2.6–4.4) 0.4 (0.3–0.8) 12.4 (11.4–12.9) 342 (300–451) 0.8 (0.5–0.9) 149 (111–235) 0.3 (0.2–0.5) 0.3 (0.2–0.4) 0.042 (0.012–0.084)	9.6 (7.8–13.3) 3.4 (2.8–4.5) 5.6 (3.0–8.1) 0.7 (0.4–1.4) 14.5 (13.2–15.9) 309 (270–358) 0.9 (0.6–1.4) 194 (118–283) 0.8 (0.4–1.1) 0.3 (0.3–0.4) 0.032 (0.028–0.033)	0.602 0.020 0.050 0.073 <0.001 0.087 0.151 0.584 0.067 0.007 0.702	0.184 0.004 0.014 0.003	0.05 0.04 0.036 0.054	0.003 0.234 <0.001 0.146

Abbreviations: FAS—fetal alcohol syndrome, Hb—hemoglobin, HLHS—hypoplastic left heart syndrome, NLR—neutrophil-to-lymphocyte ratio, SII—systemic immune inflammatory index, SIRI—systemic inflammatory response syndrome, SV—functional single ventricle.

## Data Availability

Data supporting reported results can be acquired after contact with corresponding authors for 3 years following publication after presenting justified reasons.

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
