# Peer review of "Neutrophil Count as Atrioventricular Block (AVB) Predictor following Pediatric Heart Surgery"

_ijms, 2022, doi:10.3390/ijms232012409_

Round 1

Reviewer 1 Report

In this retrospective study, the authors investigated the relationship between preoperative measurements of hematological values and the risk of post-surgical complications in the pediatric cohort, focusing in particular on the association between neutrophil counts before surgery and atrioventricular block after cardiac surgery in a cohort of 93 pediatric patients.

A relevant aspect of this work is due to the importance of the preoperative blood analysis, which is in fact a useful tool that allows a better clinical classification of the patient's disease and can be a useful strategy for patient monitoring.

Using the ROC curve, the authors defined a cut-off value for the neutrophil count, according to which it is possible to predict the peri-operative atrioventricular block.

However, there are some points that need to be clarified:

1.    The authors evaluated the relationship between low neutrophil counts and ATB. It may be of interest to evaluate the survival curve (Kaplan Meyer), in order to assess the prognostic value of the neutrophil cut-off value, by stratifying the patients' cohort based on lower or upper cut-off of the neutrophils' count.

2.    Univariate and multivariate analyzes could be useful to understand how other parameters (age, sex, pathology, lymphocyte, Hb levels, maternal autoantibody, inflammatory complication) could influence the results obtained.

3.    Since the decreased values of neutrophils seem to be due to maternal influence it is possible to report maternal autoantibody value?

4.    Minor point:

-      There are some typing mistakes throughout the manuscript to correct

Author Response

Dear Reviewer,

Thank you for your valuable comments which enabled us to improve the manuscript. We present all changes performed according to your comments.

Answering to your valuable comments: 

  1. The authors evaluated the relationship between low neutrophil counts and ATB. It may be of interest to evaluate the survival curve (Kaplan Meyer), in order to assess the prognostic value of the neutrophil cut-off value, by stratifying the patients' cohort based on lower or upper cut-off of the neutrophils' count.
  • Dear Reviewer, thank you for the comment, we added the KM curve
  1. Univariate and multivariate analyzes could be useful to understand how other parameters (age, sex, pathology, lymphocyte, Hb levels, maternal autoantibody, inflammatory complication) could influence the results obtained.
  • The univariate and multivariate analyses were performed. Age, sex, type of disease and other than Neu blood cells’ counts and Troponins did not reveal significant prognostic value for AVB.
  1. Since the decreased values of neutrophils seem to be due to maternal influence it is possible to report maternal autoantibody value?
  • Thank you for this interesting comment. However, we could not assess maternal antibodies influence since the analysis was of a retrospective manner. The idea is, however, worth of deeper analysis.

We added the relevant sentences as a limitation.

  1. Minor point:

-      There are some typing mistakes throughout the manuscript to correct       

  • Thank you, we corrected the manuscript

Kind regards

Tom U.

Reviewer 2 Report

I have received for review the paper entitled Neutrophil count as Atrioventricular block (AVB) predictor following pediatric heart surgery“. I think the article is interesting, but some changes need to be made: 

- material and method must follow the introduction, please modify

- row number 57 must be corrected

Author Response

Dear Reviewer,

Thank you for your valuable comments which enabled us to improve the manuscript. We present all changes performed according to your comments.

Answering to your valuable suggestions :

1.material and method must follow the introduction, please modify

  • Thank you for the comment, however we applied the Journal guidelines, which are slightly different than in other journals.

2. row number 57 must be corrected

Thank you for this comment, we corrected the sentence

Kind regards

Tom U.